# Towards Unsupervised Open-Set Graph Domain Adaptation via Dual Reprogramming

**Zhen Zhang**[1,2]
[1]Nanjing University
[2]National University of Singapore
zhen_zhang@nju.edu.cn

**Bingsheng He**
National University of Singapore
Singapore
dcsheb@nus.edu.sg

## Abstract

Unsupervised Graph Domain Adaptation has become a promising paradigm for transferring knowledge from a fully labeled source graph to an unlabeled target graph. Existing graph domain adaptation models primarily focus on the closed-set setting, where the source and target domains share the same label spaces. However, this assumption might not be practical in the real-world scenarios, as the target domain might include classes that are not present in the source domain. In this paper, we investigate the problem of unsupervised open-set graph domain adaptation, where the goal is to not only correctly classify target nodes into the known classes, but also recognize previously unseen node types into the unknown class. Towards this end, we propose a novel framework called GraphRTA, which conducts reprogramming on both the graph and model sides. Specifically, we reprogram the graph by modifying target graph structure and node features, which facilitates better separation of known and unknown classes. Meanwhile, we also perform model reprogramming by pruning domain-specific parameters to reduce bias towards the source graph while preserving parameters that capture transferable patterns across graphs. Additionally, we extend the classifier with an extra dimension for the unknown class, thus eliminating the need of manually specified threshold in open-set recognition. Comprehensive experiments on several public datasets demonstrate that our proposed model can achieve satisfied performance compared with recent state-of-the-art baselines. Our source codes and datasets are publicly available at https://github.com/cszhangzhen/GraphRTA.

## 1 Introduction

*Graph Neural Networks* (GNNs) have demonstrated impressive capabilities in a wide range of graph-based tasks, such as node classification [25, 14, 48], social recommendation [54, 6, 57], molecular generation [59, 34, 44] and point cloud processing [22, 43, 45], etc. Despite their great success, these GNN models often suffer from severe performance degradation when confronted with distribution shifts in graphs, such as changes in the underlying structures, node features, and label distributions [52, 30, 28]. *Unsupervised Graph Domain Adaptation* has become a promising strategy for addressing the domain shift problem by transferring knowledge across domains without relying on labels in the target domain. The majority of existing models are developed under the *closed-set* setting [62, 51, 66], which assumes that the source and target domains share the same set of classes.

However, such a strict assumption is unrealistic in real-world applications, since the target domain might introduce new classes that are absent from the source domain [64, 53], leading to significant challenges in identifying unseen samples. For instance, fraud detection models trained on previously known fraudulent behaviors from the source domain might struggle in the emerging target domain, as fraud schemes are continually evolving with fraudsters frequently developing new tactics.

39th Conference on Neural Information Processing Systems (NeurIPS 2025).

Therefore, treating all novel category instances as known classes will significantly compromise the model's ability to generalize across diverse environments. Towards this end, *open-set* graph domain adaptation [49, 33] has been proposed to accurately classify known node types into their corresponding classes while simultaneously identifying unseen node types into an unknown class. It promotes better generalization capabilities, making the model applicable across a variety of applications.

There exist some recent endeavors to explore the unsupervised open-set graph domain adaptation task [49, 40, 33, 32, 53]. The main procedure involves dividing target instances into the known and unknown groups based on their prediction entropy, then aligning the known group with the source domain. To distinguish between these two groups and enable open-set recognition, a predefined threshold is often employed, assuming that the unknown instances will exhibit higher entropy than the known ones. While promising, these methods depend heavily on manually set threshold and one threshold cannot fit all, which makes them difficult to adapt to different distributions. Additionally, they also struggle with learning clear decision boundaries, since they mainly focus on aligning source domain with the target known group, which may result in inadequate separation of the target unknown group. Hence, more efforts are necessary to tackle the challenges of open-set graph domain adaptation, particularly in recognizing and separating the target unknown group.

To address the aforementioned challenges, we propose a novel framework named GraphRTA (***Reprogram To Adapt***), which performs dual reprogramming from the graph and the model perspectives. Specifically, we reprogram the target graph by refining its structure and node features to explicitly reduce the distribution shift and encourage a clearer separation between the known and unknown groups. At the same time, we reprogram the model by pruning domain-specific parameters based on their gradients, thereby mitigating bias towards to the source graph, while retaining parameters that capture transferable patterns between the source and target graphs. Furthermore, we augment the classifier by adding an extra dimension for the unknown class, which removes the necessity for a manually specified threshold in open-set recognition. Extensive experiments across multiple public datasets indicates that our proposed model achieves superior performance in comparison to the latest state-of-the-art baselines.

In summary, our key contributions are as follows:

- We explore the challenge of unsupervised open-set graph domain adaptation, which is more practical in the real-world scenarios yet remains under-explored in the graph community.

- We are the first to reprogram both the model and the graph to improve the alignment and separation processes, offering an architecture-agnostic solution that can be applied across various GNN architectures.

- Experimental results show that GraphRTA outperforms or matches the SOTA baselines, highlighting the effectiveness of our approach in addressing the challenges associated with the open-set graph domain adaptation task.

## 2   Related Work

**Graph Neural Networks.** During the past decade, GNNs have demonstrated remarkable capability in tackling a wide range of graph learning tasks. Following the message passing framework, numerous types of GNNs have been developed, which can be broadly classified into spectral and spatial approaches [56, 50, 1]. Spectral methods, such as ChebConv [4] and Spec-GN [58], derive graph convolution operator based on spectral graph theory. In contrast, spatial models like GCN [25], GraphSAGE [14] and GAT [48] perform convolution by directly aggregating information from neighboring nodes. For comprehensive insights into these models, readers may refer to comprehensive surveys on GNNs [55, 23]. Despite their impressive performance, GNNs often depend on high-quality labeled data, which could be challenging to obtain in real-world applications. Additionally, their performance can significantly deteriorate when encountering distribution discrepancies. To mitigate this limitation, recent work has focused on adapting models trained on label-rich source domains to unlabeled target domains, thereby enhancing their generalization capabilities.

**Closed-Set Graph Domain Adaptation.** Although domain adaptation has been extensively investigated in computer vision [11, 60, 40] and natural language processing [39, 5, 61], research on graph domain adaptation remains in its early stages. Current graph domain adaptation methods primarily aim to learn domain-invariant representations, typically employing statistical matching techniques

such as maximum mean discrepancy [13] or central moment discrepancy [63], or by adopting adversarial learning mechanism [38] for implicit alignment. More specifically, UDAGCN [52] utilizes a combination of local and global graph encoders alongside adversarial training to achieve domain-invariant representations. GRADE [51] introduces a graph subtree discrepancy metric to reduce distribution shifts between source and target graphs, while SpecReg [62] applies spectral regularization to support theory-grounded graph domain adaptation. StruRW [30] proposes an edge re-weighting strategy to mitigate conditional structure shifts. A2GNN [28] highlights the inherent adaptability of graph neural networks by decoupling its transformation and propagation layers. Nevertheless, all of these aforementioned models assume that both the source and target graphs share the same label space [65, 29]. Unfortunately, this is impractical in real-world scenarios, as the target graph may contain new classes that do not exist in the source graph.

**Open-Set Graph Domain Adaptation.** Open-set domain adaptation extends closed-set domain adaptation by recognizing target novel classes that are not present in the source domain, meanwhile accurately classifying target instances that belong to the source label space [36, 41, 27, 26]. Recent models utilize a threshold to designate low-confidence samples as unknown, while aligning the source domain with the known portion of the target domain via adversarial training [19, 49, 33, 18]. Among them, DANCE [40] applies self-supervised neighborhood clustering to align each target sample with either a neighboring instance or a source prototype. PGL [33] utilizes a progressive approach to gradually reject target samples and align conditional distributions through episodic training. OpenWGL [53] introduces an uncertainty-based node representation learning framework that employs a constrained variational graph autoencoder to filter out unknown instances with a multi-sampling approach. OpenWRF [16] integrates out-of-distribution detection techniques with neighborhood information from the graph to identify novel classes. G2Pxy [64] generates both the internal and external unknown proxies via mixup to predict the distribution of novel classes in open-set learning. SDA [49] groups target representations into several clusters and employs a separate domain alignment strategy to align each target sample with either a target cluster center or a source prototype. In contrast, our model is designed from both a model-centric and a data-centric perspective, which effectively generalizes to the target domain.

# 3 The Proposed Model

## 3.1 Notations and Problem Definition

For unsupervised open-set graph domain adaptation, we are given a labeled source graph $\mathcal{G}_s = (\mathbf{X}_s, \mathbf{A}_s, \mathbf{Y}_s)$ containing $n_s$ nodes and an unlabeled target graph $\mathcal{G}_t = (\mathbf{X}_t, \mathbf{A}_t)$ with $n_t$ nodes, where the source and target graphs are sampled from different probability distributions, i.e., $\mathbb{P}(\mathcal{G}_s) \neq \mathbb{P}(\mathcal{G}_t)$. The feature matrix $\mathbf{X} \in \mathbb{R}^{n \times f}$ represents node attribute information, and the adjacency matrix $\mathbf{A} \in \mathbb{R}^{n \times n}$ indicates the connectivity information between nodes. The source graph includes a set of classes $\mathcal{C}_s$ forming the node label matrix $\mathbf{Y}_s \in \mathbb{R}^{n_s \times |\mathcal{C}_s|}$. Meanwhile, the target graph is associated with an additional set of classes $\mathcal{C}_{t \setminus s}$, collectively labeled as 'unknown', resulting in a total of $|\mathcal{C}_t| = |\mathcal{C}_s| + 1$ classes. We decompose the GNN model $\Phi(\cdot)$ into two fundamental parts: the feature extractor $f_w(\cdot)$, which transforms the graph into the node representation space, and the classifier $g_\phi(\cdot)$, which assigns class labels based on these node representations. Therefore, the GNN model $\Phi(\cdot)$ can be represented as $\Phi = f_w \circ g_\phi$. Our problem can then be formulated as follows:

*Given a graph neural network $\Phi$, a labeled source graph $\mathcal{G}_s$ with label set $\mathcal{C}_s$ and an unlabeled target graph $\mathcal{G}_t$ with label space $\mathcal{C}_t$, where $\mathcal{C}_s$ is a subset of $\mathcal{C}_t$, our goal of unsupervised graph domain adaptation is to train the model $\Phi$ using $\mathcal{G}_s$ and $\mathcal{G}_t$ to accurately classify target nodes when they belong to a label in $\mathcal{C}_s$, while marking nodes as 'unknown' if their labels fall outside of $\mathcal{C}_s$.*

## 3.2 Graph Neural Networks Revisiting

Current GNNs operate within the message passing framework [25, 14, 48], which performs convolution by iteratively aggregating representations from its local neighborhood. Taking GCN [25] as an example, the node representations at layer $l$ can be calculated as follows:

$$\mathbf{Z}^l = f_w(\mathcal{G}, \mathbf{W}^l) = \sigma(\tilde{\mathbf{D}}^{-\frac{1}{2}} \tilde{\mathbf{A}} \tilde{\mathbf{D}}^{-\frac{1}{2}} \mathbf{Z}^{l-1} \mathbf{W}^l). \tag{1}$$

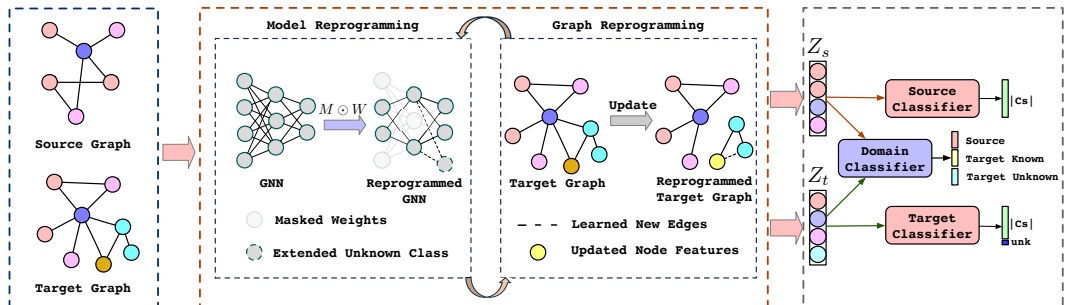

Figure 1: The pipeline of the proposed GraphRAT model. In the model reprogramming module, some nodes and connections are faded, indicating that these weights have been pruned during the reprogramming process. In the graph reprogramming module, node features are updated, and edges are dynamically modified (deleted or added), as indicated by dashed lines. Additionally, domain adversarial learning is incorporated to categorize instances into three distinct groups: source, target-known, and target-unknown, thereby enhancing the model's generalization capacity.

Here, $\sigma(\cdot)$ denotes the non-linear activation function, i.e., ReLU or LeakyReLU. The matrix $\tilde{\mathbf{A}} = \mathbf{A} + \mathbf{I}$ represents the adjacent matrix with self-connections, while $\tilde{\mathbf{D}}$ is the diagonal degree matrix of $\tilde{\mathbf{A}}$. $\mathbf{W}^l \in \mathbb{R}^{d_{l-1} \times d_l}$ indicates a matrix of trainable parameters. As demonstrated in Section 4.3, our proposed model is architecture agnostic and can be employed across diverse GNN frameworks.

## 3.3 Domain-Agnostic Model Reprogramming

We present an overview of the proposed GraphRTA framework in Figure 1. Unsupervised open-set graph domain adaptation involves training a model on a labeled source graph and an unlabeled target graph, where the target graph contains novel classes not present in the source graph. Since the model is trained using labeled data from the source graph while having no access to labels in the target graph, a primary challenge is the risk of bias toward the source domain. This discrepancy can lead the model to overemphasize source-specific patterns, thereby limiting its ability to generalize effectively to the target domain.

Motivated by the lottery ticket hypothesis [9, 2, 35, 3], which demonstrates that only a subset of parameters is crucial for generalization, we propose to reprogram the graph neural network $f_w(\cdot)$ by selectively masking its weights. Specifically, we introduce differentiable masks $\mathbf{M}^l$ to indicate the insignificant elements within the weights $\mathbf{W}^l$ of each layer. Therefore, the node representations $\mathbf{Z}$ at layer $l$ is computed as follows:

$$\mathbf{Z}^l = f_w(\mathcal{G}, \tilde{\mathbf{W}}^l) = \sigma(\tilde{\mathbf{D}}^{-\frac{1}{2}} \tilde{\mathbf{A}} \tilde{\mathbf{D}}^{-\frac{1}{2}} \mathbf{Z}^{l-1} (\mathbf{W}^l \odot \mathbf{M}^l)), \tag{2}$$

where $\odot$ denotes the element-wise product. Hyperparameter $\rho$ is utilized to control the sparsity of the masks, determining the proportion of weights to be retained. To quantify the importance of each weight element, we calculate its gradient $\nabla \mathbf{M}^l$ with respect to the loss function. The absolute value of these gradients is then used as the importance score, reflecting how much each weight contributes to generalization in the target domain. Weights with smaller gradients typically contribute less to reducing the loss and may capture domain-specific patterns that do not help in adapting to the target domain and recognition of unseen classes. By masking such weights, the reprogrammed model can focus better on transferable features that generalize across domains. Hence, we set the lowest $\rho$ percent of gradient values in $\mathbf{M}^l$ to zero, leaving the remaining elements at 1. These sparse masks are then applied to prune $\mathbf{W}^l$, resulting in the reprogrammed sparse model.

To handle open-set recognition, we further reprogram the output layer by augmenting an extra dimension for the unknown class as follows:

$$g_\phi(\mathbf{z}) = [\phi^\top \mathbf{z}, \hat{\mathbf{w}}^\top \mathbf{z}], \tag{3}$$

where $\phi \in \mathbb{R}^{d \times |\mathcal{C}_s|}$ represents the closed-set classifier and $\hat{\mathbf{w}} \in \mathbb{R}^{d \times 1}$ denotes the linear projection layer for the unknown class. The augmented logits are then passed through a softmax layer to generate the posterior probabilities, with the final prediction assigned to the class with the highest probability. This mechanism is designed to effectively distinguish between known and unknown

classes through using a dynamic threshold based on the input node representation $\mathbf{z}$, whereas existing models typically rely on a manually defined threshold, making them less adaptable to domain discrepancies. Through these model reprogramming procedures, we develop a domain-agnostic GNN model capable of identifying open-set classes.

## 3.4  Distribution-Aware Graph Reprogramming

While our proposed model reprogramming module can help mitigate source bias, it does not address the fundamental domain shift that arises from the structural and feature differences between the source and target graph data. Existing approaches often overlook the fact that the domain shift is inherently caused by the input graph's unique characteristics. This discrepancy makes it challenging for the GNN model to generalize well across domains in the open-set scenario, as the target graph may contain novel structures, features or classes that the source domain does not encompass.

To overcome this limitation, we further propose performing graph reprogramming, where the target graph itself is refined to improve compatibility between the source and target domains. Through this approach, we modify the target graph's structure and node features to better align with the source domain, while differentiating the known and unknown groups within the target domain [66]. This strategy not only mitigates domain shift but also strengthens the model's ability to generalize and recognize the unseen classes. More specifically, we implement graph reprogramming by applying transformation functions to adjust both the target graph structure and node features:

$$\hat{\mathbf{X}}_t = \psi_x(\mathbf{X}_t), \ \hat{\mathbf{A}}_t = \psi_a(\mathbf{A}_t), \tag{4}$$

where $\psi_x(\cdot)$ represents the transformation function applied to update node features, and $\psi_a(\cdot)$ denotes the function for modifying the graph structure by adding or removing edges. Although numerous approaches, like graph structure learning methods [7, 21, 31], can be employed to adjust the graph data, we choose two simple, direct transformation strategies described below, with additional options explored in the ablation study in Appendix B.

For node features, we define the transformation function $\psi_x(\mathbf{X}_t) = \mathbf{X}_t + \Delta\mathbf{X}_t$, where $\Delta\mathbf{X}_t \in \mathbb{R}^{n_t \times f}$ represents a set of continuous, learnable parameters. This simple formulation allows for either masking node features (setting them to zero) or modifying their values, enhancing the models flexibility in refining node representations. For graph structure, the transformation function $\psi_a(\cdot)$ is modeled as $\psi_a(\mathbf{A_t}) = \mathbf{A}_t \oplus \Delta\mathbf{A}_t$, where $\Delta\mathbf{A}_t \in \mathbb{R}^{n_t \times n_t}$ is a binary matrix that adjusts the graph's structure by adding or removing edges. The operation $\oplus$ denotes element-wise exclusive OR (XOR), where if both the corresponding values in $\mathbf{A}_t$ and $\Delta\mathbf{A}_t$ are 1, the XOR operation results in 0, effectively deleting the edge. Conversely, if one value is 0 and the other is 1, an edge is added. To regulate the changes to the graph structure, we impose a constraint on the total number of modifications made to the adjacency matrix. Particularly, the sum of the entries in $\Delta\mathbf{A}_t$ is limited by a pre-defined budget $\mathcal{B}$, i.e., $\sum \Delta\mathbf{A}_t \leq \mathcal{B}$, preventing excessive changes that might deviate too much from the original graph and ensuring computational efficiency. Through these graph reprogramming processes, we dynamically update the target graph to mitigate distribution shifts and facilitate open-set recognition.

## 3.5  Training Procedure

In this section, we detail the process of updating the model reprogramming and graph reprogramming modules using the proposed losses. Following previous works [12, 20, 33, 46], we adopt a domain adversarial learning framework to match the source and target distributions through feature alignment. In an open-set scenario, where the target domain contains unknown classes not present in the source domain, such alignment approach can cause negative transfer due to class set mismatches. Existing methods focus on aligning source and target features within the known class set, ignoring any alignment signal from target instances that belong to unknown classes. As a result, the classifier is unable to establish a clear decision boundary for unknown classes, as target-unknown instances remain entangled with known ones in the aligned feature space.

To this end, we aim at explicitly pushing target-unknown features apart from both the source and target-known features, while ensuring alignment between source and target-known features. Specifically, we first calculate the entropy value for each target instance based on the known classes as

follows:

$$e_i = \mathcal{H}(\mathbf{p}_i) = -\sum_{k=1}^{|\mathcal{C}_s|} \mathbf{p}_{ik}\log(\mathbf{p}_{ik}), \tag{5}$$

where $\mathbf{p}_i$ is the output probability generated by the classifier $g_\phi(\cdot)$. The entropy value, which quantifies the uncertainty, serves as an indicator for open-set recognition, where a high entropy suggests the instance may belong to the unknown class. After normalizing the entropy values, we model them as being generated by a mixture of two Beta distributions to capture the overall characteristics of the target graph [20]:

$$p(e_i) = \mu_{tk} \cdot p(e_i|tk) + \mu_{tu} \cdot p(e_i|tu), \tag{6}$$

where $e_i$ represents the entropy value for node $v_i$. $p(e_i|tk)$ and $p(e_i|tu)$ denote the probability density functions for the target-known (*tk*) and target-unknown groups (*tu*), respectively. Meanwhile, $\mu_{tk}$ and $\mu_{tu}$ are the mixing coefficients for these two distributions. Then, we perform posterior inference by fitting a Beta mixture model using the Expectation-Maximization (EM) algorithm:

$$p(tk|e_i) = \frac{\mu_{tk} \cdot p(e_i|tk)}{\mu_{tk} \cdot p(e_i|tk) + \mu_{tu} \cdot p(e_i|tu)}. \tag{7}$$

Thus, we estimate the probability that an instance belongs to the target-known group based on its entropy value without using any thresholds and $p(tu|e_i) = 1 - p(tk|e_i)$.

After estimating the probability of each target instance belonging to either the target-known or target-unknown group, we can classify all the instances into three distinct domains for the domain adversarial learning framework, i.e., *source*, *target-known*, and *target-unknown*. To achieve this goal, we introduce a domain discriminator $d_\theta(\cdot)$, implemented as a multi-layer perceptron (MLP), which engages in a minimax game with the feature extractor $f_w(\cdot)$. The feature extractor works to learn node representations that make it challenging for the discriminator to identify the origin of each node. We implement adversarial training using a Gradient Reversal Layer (GRL) [12], which promotes the maximization of feature representations. Meanwhile, the domain discriminator $d_\theta(\cdot)$ is optimized by minimizing the cross-entropy loss to effectively classify domains as follows:

$$\mathcal{L}_{adv} = -\frac{1}{n_s + n_t} \sum_{i=1}^{n_s+n_t} \sum_{k=1}^{3} \mathbf{y}_{ik}\log(\hat{\mathbf{y}}_{ik}), \tag{8}$$

where $\mathbf{y}_i = [1, 0, 0]$ when the node is from the source graph, and $\mathbf{y}_i = [0, p(tk|e_i), p(tu|e_i)]$ when the node is from the target graph. $\hat{\mathbf{y}}_i$ represents the domain prediction for node $v_i$. Thus, the adversarial learning loss simultaneously aligns and segregates the three sets to learn domain-invariant representations.

Furthermore, we update the model reprogramming module by utilizing the label information in the source graph as follows:

$$\mathcal{L}_{cls} = \sum_{i=1}^{n_s} \mathcal{L}_{ce}(\Phi(\mathbf{x}_i), y_i) + \lambda\mathcal{L}_{ce}(\Phi(\mathbf{x}_i)\backslash y_i, |\mathcal{C}_s| + 1), \tag{9}$$

where $\mathcal{L}_{ce}$ represents the cross-entropy loss, while $\Phi(\cdot)$ denotes the reprogrammed model. $\lambda$ is a trade-off hyper-parameter. The first term focuses on optimizing the augmented output to match the ground truth labels, thereby preserving performance on the closed set. For the second term, $\Phi(\mathbf{x}_i)\backslash y_i$ removes the probability associated with the ground truth label and aligns it with class $|\mathcal{C}_s|+1$, which can be regarded as a simplified mixup approach. By masking out the ground truth, we ensure that the model is explicitly trained to classify instances as unknown when they do not align with any of the known class patterns. For graph reprogramming module, we incorporate entropy minimization loss to encourage confident predictions for the unlabeled target instances, while simultaneously distinguishing target-known features from target-unknown features as follows:

$$\mathcal{L}_{ent} = \sum_{i=1}^{n_t} \mathcal{H}(\Phi(\mathbf{x}_i)) + \mathcal{L}_{ce}(\hat{\mathbf{y}}_i, p(tu|e_i)), \tag{10}$$

where $\mathcal{H}(\cdot)$ is the entropy function defined in Eq.( 5), while minimizing $\mathcal{L}_{ce}(\hat{\mathbf{y}}_i, p(tu|e_i))$ enables the graph to generate discriminative features specifically for target-unknown instances. Therefore, the overall loss function is:

$$\mathcal{L} = \mathcal{L}_{adv} + \mathcal{L}_{cls} + \mathcal{L}_{ent}. \tag{11}$$

# 4 Experiments

## 4.1 Experimental Settings

**Datasets.** To thoroughly assess the performance of our proposed GraphRTA, we conduct experiments using three categories of publicly available datasets. An overview of these dataset characteristics is provided in Table 1, with more detailed descriptions as follows.

The first category comprises three *citation* datasets, i.e., DBLPv7*(D)*, Citationv1*(C)*, and ACMv9*(A)* [28], where nodes represent individual papers, while edges indicate citation relationships. Particularly, DBLPv7 encompasses DBLP papers published between 2004

Table 1: Dataset statistics.

| Datasets | #Nodes | #Edges | #Feat | #Class |
|----------|--------|--------|-------|--------|
| DBLPv7 | 5,484 | 8,117 | | |
| Citationv1 | 8,935 | 15,098 | 6,775 | 5 |
| ACMv9 | 9,360 | 15,556 | | |
| ogbn-arxiv | 169,343 | 1,166,243 | 128 | 40 |
| Cornell | 183 | 298 | | |
| Texas | 183 | 325 | 1,703 | 5 |
| Wisconsin | 251 | 515 | | |

and 2008, Citationv1 contains articles from Microsoft Academic Graph up to 2008, and ACMv9 consists of papers published by ACM from 2000 to 2010. Each paper is classified into one of the five distinct research topics: Databases, Artificial Intelligence, Computer Vision, Information Security, and Networking.

We further include the ***ogbn-arxiv*** dataset [17], which is composed of computer science papers from arXiv. Each paper is represented by a 128-dimensional feature vector, derived by averaging the embeddings of the words in its title and abstract. We partition this dataset into three temporal domains according to the publication years: 1950-2016 *(Arxiv I)*, 2016-2018 *(Arxiv II)*, and 2018-2020 *(Arxiv III)*. The task involves classifying each paper into one of 40 predefined subject areas under the temporal shifts.

Lastly, we incorporate the ***WebKB*** dataset [37], a webpage dataset from computer science departments across various universities. Among them, we select three heterophily graphs (i.e., ***Cornell***, ***Texas***, and ***Wisconsin***), where nodes represent web pages and edges indicate hyperlinks between them. Each webpage is denoted by bag-of-words features, then we categorize them into the following five groups: student, project, course, staff, or faculty.

**Baselines.** We compare our proposed GraphRTA against a broad set of recent baselines across three key categories. *(1) Graph Neural Networks*: This category includes traditional GNN models like GCN [25], SAGE [14], and GAT [48]. They are trained on the source graph and then evaluated directly on the target graph without adaptations between domains. For open-set recognition, a threshold is applied to identify instances belonging to unseen categories. *(2) Closed-Set Graph Domain Adaptation*: Approaches in this group focus on graph domain adaptation within the closed-set settings, where the source and target graphs share the same label space. We compare our model with several recent methods including UDAGCN [52], GRADE [51], SpecReg [62], StruRW [30], and A2GNN [28]. Similarly, we employ a predefined threshold to identify open-set instances. *(3) Open-Set Graph Learning or Domain Adaptation*: This group of methods is designed for scenarios where the target graph contains categories that do not exist in the source graph. We consider DANCE [40], OpenWGL [53], PGL [33], OpenWRF [16], G2Pxy [64], SDA [49] and UAGA [42] for comparisons. These approaches are strong baselines for assessing our model's capability to transfer knowledge to the target domain and generalize effectively to unseen categories.

**Implementation Details.** In this work, we adopt the experimental setup used in prior researches [64, 49], where a portion of classes is reserved as "unknown" with the remaining classes regarded as "known". Specifically, $|\mathcal{C}_s|$ is 3 in Citation and WebKB datasets, and 30 for the ogbn-arxiv dataset. The domain adaptation models are trained using labeled source nodes from the known classes along with unlabeled nodes from the target graph. Among them, 70% of the labeled source nodes are utilized for training, 10% are set aside for validation, and the remaining 20% serve as a sanity check. The final evaluation is conducted on the target nodes. For a fair comparison, we utilize the baselines' publicly available source codes and tune their hyperparameters to their optimal values using the validation set. Our proposed GraphRTA is implemented using PyTorch Geometric [8] and optimized with the Adam optimizer [24]. Hyperparameters for learning rate, weight decay and $\lambda$ are searched within the ranges of $[0.1, 0.01, 0.001, 1e^{-4}, 1e^{-5}]$, and the sparse constraint $\rho$ is explored within the interval $[0, 1]$. The experiments are repeated five times, and performance metrics are reported as the

Table 2: Node classification accuracy and H-score (mean ± std) for citation datasets. The best results are shown in bold with the second-best results underlined.

| Models | ACMv9→Citationv1 | | ACMv9→DBLPv7 | | Citationv1→ACMv9 | | Citationv1→DBLPv7 | | DBLPv7→ACMv9 | | DBLPv7→Citationv1 | |
|---|---|---|---|---|---|---|---|---|---|---|---|---|
| | Acc | HS | Acc | HS | Acc | HS | Acc | HS | Acc | HS | Acc | HS |
| GCN | 40.64±0.98 | 41.02±2.11 | 45.84±1.06 | 50.20±1.26 | 47.10±0.49 | 49.05±0.74 | 51.48±0.42 | 56.13±0.22 | 43.90±1.50 | 44.47±2.80 | 39.26±0.70 | 37.96±1.34 |
| SAGE | 38.24±0.80 | 36.89±2.10 | 41.77±1.05 | 45.20±1.41 | 43.90±1.56 | 45.63±2.18 | 47.14±0.82 | 51.65±0.83 | 41.66±0.47 | 42.04±1.32 | 39.62±0.22 | 40.32±0.40 |
| GAT | 32.01±0.73 | 21.51±2.36 | 34.84±1.25 | 32.90±2.66 | 36.38±0.66 | 30.67±1.66 | 34.56±0.64 | 32.22±1.53 | 35.85±1.91 | 25.66±5.30 | 32.87±1.41 | 20.91±3.92 |
| UDAGCN | 44.78±4.12 | 20.94±6.21 | 55.07±1.03 | 50.05±7.90 | 53.38±1.53 | 53.56±6.00 | 62.36±4.19 | 43.21±4.08 | 47.28±1.49 | 39.21±1.16 | 52.38±1.18 | 46.92±6.26 |
| GRADE | 57.23±1.06 | 59.49±1.16 | 56.12±0.65 | 58.14±1.07 | 57.86±0.29 | 60.41±0.26 | 61.94±0.38 | 64.21±0.48 | 54.93±0.40 | 57.73±0.41 | 54.60±0.64 | 57.36±0.55 |
| SpecReg | 51.31±5.60 | 36.70±1.49 | 58.17±2.06 | 60.15±0.41 | 56.58±1.22 | 56.36±0.86 | 63.68±5.82 | 59.62±0.59 | 53.30±4.05 | 53.12±6.84 | 55.72±2.43 | 49.43±7.41 |
| StruRW | 46.47±4.63 | 42.36±4.25 | 46.91±1.95 | 46.08±3.60 | 51.91±0.60 | 45.38±0.13 | 56.19±0.10 | 54.08±1.19 | 48.86±0.62 | 43.85±1.74 | 51.08±0.91 | 43.25±1.29 |
| A2GNN | 42.53±2.07 | 41.66±3.69 | 60.43±0.52 | 62.74±0.82 | 57.21±1.03 | 57.12±0.63 | 63.45±0.31 | 65.37±0.59 | 57.64±1.82 | 60.68±2.27 | 41.09±1.20 | 43.52±1.24 |
| DANCE | 57.77±0.64 | 60.94±0.76 | 58.01±0.47 | 61.31±0.62 | 58.76±0.36 | 61.33±0.55 | 62.97±0.65 | 65.42±1.20 | 55.97±0.62 | 58.90±0.60 | 55.77±0.56 | 58.95±0.70 |
| OpenWGL | 49.98±0.62 | 5.57±1.56 | 52.43±0.62 | 7.86±0.69 | 48.37±0.50 | 4.07±0.99 | 55.68±0.35 | 3.49±0.76 | 48.04±1.08 | 25.13±4.89 | 49.52±0.68 | 22.05±2.29 |
| PGL | 54.42±1.04 | 57.86±1.12 | 48.43±1.12 | 53.15±1.23 | 51.87±0.69 | 54.71±0.72 | 53.83±0.66 | 59.01±0.75 | 49.27±0.80 | 51.74±0.83 | 52.82±0.87 | 56.16±0.93 |
| OpenWRF | 53.53±2.59 | 31.05±4.57 | 48.16±1.63 | 35.45±3.62 | 47.01±3.32 | 33.08±6.46 | 52.81±1.43 | 31.96±1.24 | 47.27±1.54 | 19.38±4.03 | 57.31±1.14 | 34.06±9.77 |
| G2Pxy | 59.75±0.49 | 54.47±1.33 | 56.26±0.73 | 49.63±2.41 | 58.49±1.03 | 58.56±1.36 | 61.42±0.47 | 59.13±0.54 | 54.48±0.58 | 54.82±0.78 | 56.36±0.69 | 54.02±1.75 |
| SDA | 58.23±4.67 | 59.97±6.74 | 59.06±4.75 | 56.34±1.23 | 57.33±6.22 | 58.85±8.58 | 63.55±0.91 | 65.53±2.00 | 57.66±0.61 | 60.54±1.20 | 57.27±2.72 | 59.73±3.87 |
| UAGA | 53.37±6.72 | 61.34±1.16 | 52.11±2.96 | 67.50±2.95 | 52.25±4.81 | 60.59±5.95 | 52.73±5.13 | 64.81±4.50 | 48.14±1.47 | 55.73±3.98 | 47.97±9.24 | 52.16±2.18 |
| GraphRTA | 66.26±0.93 | 66.33±1.69 | 62.33±0.68 | 64.42±1.10 | 60.93±2.63 | 62.89±2.46 | 63.87±1.97 | 65.99±1.87 | 56.91±2.50 | 59.41±2.22 | 60.11±1.98 | 62.33±1.53 |

Table 3: Node classification accuracy and H-score (mean ± std) for ogbn-arxiv and WebKB datasets. OOM means out-of-memory. '-' indicates cases where baseline methods encounter errors because their predefined strategies are not satisfied.

| Models | Arxiv I→Arxiv II | | Arxiv I→Arxiv III | | Arxiv II→Arxiv III | | Cornell→Wisconsin | | Texas→Cornell | | Texas→Wisconsin | |
|---|---|---|---|---|---|---|---|---|---|---|---|---|
| | Acc | HS | Acc | HS | Acc | HS | Acc | HS | Acc | HS | Acc | HS |
| GCN | 44.82±0.20 | 41.08±0.74 | 41.57±0.22 | 41.05±0.66 | 45.65±0.37 | 41.04±0.89 | 21.19±0.17 | 0.20±0.45 | 38.46±8.43 | 25.19±11.00 | 21.83±6.29 | 13.32±9.15 |
| SAGE | 44.95±0.15 | 37.83±0.71 | 42.75±0.15 | 38.63±0.64 | 49.60±0.12 | 38.11±0.49 | 18.56±2.86 | 10.30±10.7 | 34.75±3.31 | 11.66±4.91 | 24.22±10.7 | 14.48±9.01 |
| GAT | 44.81±0.13 | 34.31±1.39 | 42.05±0.30 | 34.97±0.76 | 46.49±0.17 | 36.35±0.78 | 20.96±0.21 | 0.40±0.54 | 27.97±3.00 | 15.89±9.15 | 9.48±2.31 | 8.08±2.24 |
| UDAGCN | 31.90±2.27 | 33.68±3.48 | 27.71±0.86 | 28.37±1.63 | 35.09±1.29 | 39.53±1.56 | 19.28±6.57 | 5.94±5.51 | 29.18±1.87 | 7.35±5.28 | 22.78±5.19 | 3.93±4.12 |
| GRADE | 43.01±0.20 | 47.18±0.29 | 39.28±0.37 | 44.19±0.41 | 42.80±0.17 | 46.09±0.21 | 17.05±11.5 | 12.52±1.26 | 28.96±7.54 | 21.52±11.5 | 24.46±7.32 | 23.84±4.67 |
| SpecReg | 37.80±1.90 | 31.76±1.64 | 28.14±4.59 | 29.03±3.87 | 46.60±0.29 | 31.70±4.27 | 20.79±3.24 | 19.93±3.12 | 31.69±3.34 | 13.53±9.15 | 19.20±4.72 | 10.28±6.14 |
| StruRW | 37.47±1.93 | 40.67±2.09 | 36.17±0.27 | 40.54±0.45 | 42.10±0.44 | 43.62±0.50 | 16.57±2.26 | 16.22±3.72 | 42.02±7.51 | 40.98±7.86 | 16.01±2.47 | 11.46±6.17 |
| A2GNN | 42.07±0.14 | 45.00±0.24 | 38.92±0.16 | 43.14±0.17 | 42.26±0.53 | 45.18±0.17 | 19.12±2.56 | 17.40±3.32 | 44.37±0.71 | 31.59±6.44 | 14.98±0.77 | 6.22±2.58 |
| DANCE | OOM | OOM | OOM | OOM | OOM | OOM | 21.52±23.4 | 3.11±0.49 | 20.77±0.00 | 0.00±0.00 | 4.22±0.21 | 1.65±1.50 |
| OpenWGL | 32.58±0.58 | 1.45±0.40 | 32.99±1.49 | 1.31±0.37 | 35.46±2.61 | 0.04±0.08 | 16.57±2.49 | 14.09±2.30 | 33.22±5.77 | 28.99±5.65 | 10.51±4.80 | 8.76±3.56 |
| PGL | 41.50±0.25 | 46.32±0.28 | 38.38±0.14 | 43.31±0.16 | 40.28±0.24 | 45.43±0.27 | 18.24±7.16 | 0.00±0.00 | 21.20±0.45 | 0.00±0.00 | 28.45±5.93 | 0.00±0.00 |
| OpenWRF | 32.58±0.58 | 1.45±0.40 | 32.99±1.49 | 1.31±0.37 | 35.46±2.61 | 0.04±0.08 | 26.29±7.11 | 6.84±5.81 | 21.64±3.81 | 4.58±5.03 | 26.93±11.4 | 5.99±4.65 |
| G2Pxy | 31.13±4.55 | 28.45±1.13 | 24.79±1.49 | 16.28±3.25 | 34.93±3.03 | 37.27±6.75 | - | - | - | - | - | - |
| SDA | 39.77±0.34 | 42.60±0.15 | 36.37±0.21 | 39.44±0.21 | 41.53±0.20 | 46.03±0.18 | - | - | - | - | - | - |
| UAGA | 32.92±0.16 | 23.69±0.16 | 32.24±0.13 | 22.98±0.13 | 39.16±0.39 | 29.79±0.41 | - | - | - | - | - | - |
| GraphRTA | 47.70±1.39 | 50.79±2.79 | 45.52±2.00 | 46.25±0.40 | 52.37±1.49 | 48.42±1.94 | 33.46±4.43 | 34.36±1.76 | 52.18±0.38 | 35.64±0.83 | 29.34±1.21 | 30.08±2.96 |

mean along with standard deviations for both accuracy and H-score [10]. The H-score combines the accuracies of the target known and target unknown classes to provide a balanced assessment of the model's performance as follows:

$$HS = \frac{2 \times Acc_{tk} \times Acc_{tu}}{Acc_{tk} + Acc_{tu}}, \tag{12}$$

where $Acc_{tk}$ denotes accuracy on the target known classes, and $Acc_{tu}$ indicates accuracy on the target unknown classes. A higher score means balanced performance across known and unknown target categories, offering a more comprehensive evaluation metric.

## 4.2 Results and Analyses

We present the overall results in Table 2 and Table 3. As we can see, our proposed GraphRTA consistently demonstrates superior performance across a variety of scenarios. Within the three baseline categories, standard GNN models show limited performance, due to their lack of mechanisms to address distribution shifts between source and target graphs. Closed-set adaptation baselines, which account for these distribution shifts, yield better results than standard GNNs but struggle to handle open-set classes effectively. In contrast, open-set baselines generally perform well by incorporating strategies to recognize and manage previously unseen classes in the target graph, though they are still outperformed by GraphRTA in most cases.

We observe that the H-score provides a more comprehensive evaluation metric than accuracy. Several baselines achieve high accuracy scores but suffer from low H-scores, reflecting their difficulty in accurately identifying open-set instances. For example, while OpenWGL achieves an accuracy of 49.98% in the scenario of ACMv9→Citationv1, its H-score is only about 5.57%, illustrating its limitations in open-set recognition. Additionally, most models struggle with heterophilous datasets like ogbn-arxiv and WebKB, which pose additional challenges due to their weakly correlated characteristics. Two recent baselines encounter issues in this context, for instance, G2Pxy fails to meet

Table 4: Classification H-score with different known classes.

| Classes | A→C | A→D | C→A | C→D | D→A | D→C |
|---|---|---|---|---|---|---|
| $C_{\text{kwn}} = 2$ | 53.39±1.87 | 45.29±3.46 | 41.10±2.63 | 41.54±2.92 | 44.48±2.86 | 54.18±4.41 |
| $C_{\text{kwn}} = 3$ | 66.33±1.69 | 64.42±1.10 | 62.89±2.46 | 65.99±1.87 | 59.41±2.22 | 62.33±1.53 |
| $C_{\text{kwn}} = 4$ | 65.74±0.54 | 63.28±0.69 | 61.63±0.64 | 67.20±0.63 | 57.12±0.35 | 59.82±0.69 |

Table 5: Classification H-score with different openset class detection strategies.

| Methods | A→C | A→D | C→A | C→D | D→A | D→C |
|---|---|---|---|---|---|---|
| GraphRTA$_{\text{thres}}$ | 63.07±0.13 | 61.21±1.07 | 61.00±1.52 | 63.89±1.35 | 58.88±0.72 | 59.73±1.10 |
| GraphRTA | 66.33±1.69 | 64.42±1.10 | 62.89±2.46 | 65.99±1.87 | 59.41±2.22 | 62.33±1.53 |

the predefined rules for node embedding mixup, while SDA struggles to form meaningful clusters under these conditions. In contrast, GraphRTA overcomes these limitations by leveraging model and graph reprogramming, enabling it to deliver robust performance without such constraints.

### 4.3 Ablation Studies

**Impact of Different Known Classes.** We have also conducted additional ablation studies to evaluate the impact of varying the number of known classes on the model's performance. Specifically, we aim to understand how the size of the known class subset influences the model's ability to discriminate between known and unknown categories. As the number of known classes increases, the boundary between known and unknown nodes becomes more complex, potentially affecting the generalization capability of the domain adaptation process. The results, reported in terms of the H-score on the citation dataset, are presented in Table 4. When the number of known classes is very small (e.g., only 2 known classes), open-set models have limited supervision to guide effective feature alignment and decision boundary formation, leading to less stable performance. In contrast, as more known classes are introduced, the model benefits from richer supervision, enabling more discriminative representations and improving robustness against the presence of unknown classes. These observations highlight that our model becomes progressively more reliable and less sensitive to open-set uncertainty as the number of known classes increases.

**Alternative Open-set Detection Strategies.** We further implemented an entropy-based thresholding variant, which replaces the 3-class classification scheme with a confidence-based decision rule. Specifically, samples with prediction entropy exceeding a pre-defined threshold are regarded as belonging to unknown classes. We then compared this variant with our proposed approach across all experimental settings. The results, measured by the H-score in Table 5, show that our method consistently outperforms the entropy-based thresholding strategy, demonstrating its superior ability to distinguish between known and unknown nodes in the open-set graph domain adaptation scenario.

**Impact of Different GNN Architectures.**
As discussed in Section 3.2, our proposed GraphRTA is designed to be compatible with a range of GNN architectures, allowing flexibility across models. We examine its performance using 3 prominent GNN frameworks: GCN [25], GraphSAGE [14], and GAT [48], with results presented in Ta-

Table 6: H-scores with different GNN architectures.

| Architectures | D→C | D→A | A→C |
|---|---|---|---|
| GraphRTA$_{\text{GCN}}$ | 62.33±1.53 | 59.41±2.22 | 66.33±1.69 |
| GraphRTA$_{\text{SAGE}}$ | 59.09±2.66 | 55.45±1.49 | 63.11±1.48 |
| GraphRTA$_{\text{GAT}}$ | 54.17±6.60 | 51.82±9.14 | 57.51±6.53 |

ble 6. These results indicate that while all architectures benefit from the proposed dual reprogramming approach, their performance varies across different datasets, highlighting the influence of architectural design on adaptation capabilities. Among them, GAT achieves the lowest performance, which adapt poorly from the source to target graphs due to its attention mechanism. The multi-head attention paradigm also requires substantial parameter tuning, which may hinder adaptation. Interestingly, GCN, despite its simpler design, consistently performs well, demonstrating resilience across most tasks.

**Visualization.** To better understand the quality of the learned node representations, we use t-SNE [47] to project them into a 2-D space and visualize the results in Figure 2 through scatter plots. More specifically, the three known classes are represented by red, blue, and green, while the unknown class is depicted in orange. The vanilla GCN [25], without any adaptation, struggles to

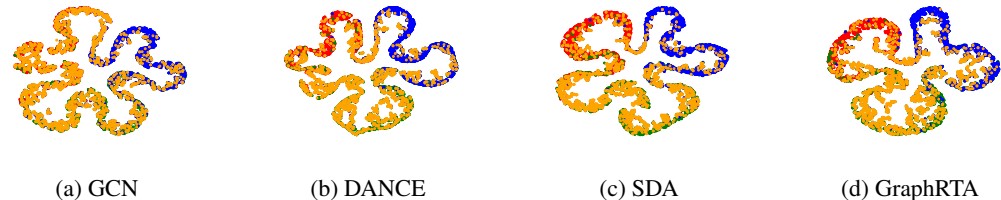

| (a) GCN | (b) DANCE | (c) SDA | (d) GraphRTA |

Figure 2: Visualization of node representations in the target graph for the citation dataset (A→C) with the unknown class highlighted in orange.

produce distinct clusters, leading to significant overlap between nodes of the unknown class and those of the known classes. This overlap occurs because the model lacks mechanisms to address distribution shifts between the source and target graphs. In contrast, two representative open-set baselines, DANCE [40] and SDA [49], are able to identify nodes from the target unknown class, but their boundaries are blurred, with most nodes from unknown classes often blending into known class clusters. Our proposed GraphRTA achieves relatively clearer separation, generating compact clusters for known classes while effectively isolating open-set instances.

**Hyper-parameter Sensitivity.** In this section, we analyze the effects of two important hyperparameters: the sparsity ratio $\rho$ within the model reprogramming module and the structural modification ratio $\mathcal{B}$ in the graph reprogramming module. As illustrated in Figure 3, the sparsity ratio significantly influences the model's performance; as the masking ratio increases, the performance declines sharply. This impairs the model's

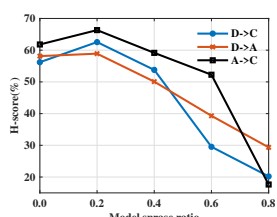
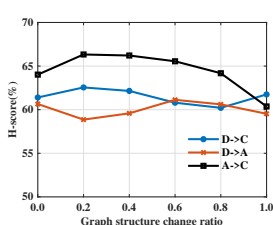

(a) Model reprogramming ratio  (b) Graph reprogramming ratio

Figure 3: H-scores under different ratios in reprogramming.

ability to capture essential patterns due to excessive weight masking. In contrast, the performance impact of adjusting the graph structure change ratio is relatively robust, suggesting that the model can adapt to moderate structural alterations in the graph without significant degradation. Additional analyses of other hyperparameters and representation visualizations are provided in the Appendix B.

## 5   Conclusion

This paper studies unsupervised open-set graph domain adaptation, an under-explored area in the graph community, where the target graph introduces new classes that are not present in the source graph. To address the source bias and distributional shift problems, we propose a novel framework named GraphRTA that conducts dual reprogramming at the model as well as the graph levels. Through extensive evaluations on a variety of public datasets, we further show that our proposed GraphRTA consistently outperforms or matches the performance of recent state-of-the-art models. In future work, we aim to extend our framework to address additional challenges in graph adaptation, such as source-free open-set graph domain adaptation, semi-supervised open-set graph domain adaptation, and out-of-distribution detection, thereby broadening its applicability and enhancing its robustness as well as generalization in real-world scenarios.

## 6   Border Impacts and Limitations

This paper advances the field of machine learning, particularly in Open-Set Graph Domain Adaptation, which enables graph models to adapt to new, unseen classes across domains. It has broad applications, including fraud detection, biological network analysis, and recommendation systems. While our work improves model robustness and generalization, potential societal impacts include both benefits (e.g., better adaptability in real-world graph-based systems) and challenges (e.g., risks of biased adaptation or misclassification in high-stakes applications). However, we do not identify any immediate or specific societal concerns.

## Acknowledgments and Disclosure of Funding

This research is supported by the National Research Foundation, Singapore and Infocomm Media Development Authority under its Trust Tech Funding Initiative. Any opinions, findings and conclusions or recommendations expressed in this material are those of the author(s) and do not reflect the views of National Research Foundation, Singapore and Infocomm Media Development Authority.

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

Figure 4: Accuracy under different layers and dimension.

Table 7: Classification H-score with different components.

| Architectures | D→C | D→A | A→C |
|---|---|---|---|
| GraphRTA$_{\text{w/o MR}}$ | 59.86±0.38 | 56.31±0.68 | 64.93±0.24 |
| GraphRTA$_{\text{w/o GR}}$ | 60.34±0.62 | 56.71±1.29 | 62.95±1.97 |
| GraphRTA | 62.33±1.53 | 56.91±2.50 | 66.33±1.69 |

## A  Running Environment

Our experiments are conducted on a Linux server with 2 AMD EPYC 7543 CPU@2.80GHz, 512G RAM and one NVIDIA A100-SXM4-80GB GPU. The proposed model is implemented with Pytorch 1.13.1 in Python 3.8 using Pytorch Geometric 2.4.0.

## B  Additional Ablation Studies and Analyses

In this section, we conduct a series of ablation studies to comprehensively assess the effectiveness of our proposed GraphRTA framework.

**Sensitivity to Two Key Hyper-Parameters.** We begin by analyzing the model's sensitivity to two critical hyper-parameters: the number of layers $L$ and the node representation dimension $d$. As shown in Figure 4, model performance initially improves with an increase in the number of layers but starts to degrade beyond a certain point (i.e., 2 layers). This decline is attributed to overfitting, as deeper layers may overly adapt to the training data while failing to generalize effectively. In contrast, the model exhibits consistent robustness to variations in the node representation dimension, highlighting its ability to perform well across a range of dimensional configurations.

**Effectiveness of Model and Graph Reprogramming.** Next, we further investigate the individual contributions of the model reprogramming (i.e., MR) and graph reprogramming (i.e., GR) components. Results presented in Table 7 provide insights into the impact of excluding these components. Specifically, the configuration labeled as "w/o MR" excludes the model reprogramming module, while "w/o GR" omits the graph reprogramming module. A configuration without either component demonstrates the most significant performance degradation, underscoring the necessity of incorporating both modules. These findings highlight that the dual reprogramming strategy is critical for mitigating domain shifts and improving model adaptability.

Table 8: H-score with different graph reprogramming strategies.

| Architectures | D→C | D→A |
|---|---|---|
| GraphRTA$_{\text{w SUBLIME}}$ [31] | 59.72±1.03 | 54.28±1.48 |
| GraphRTA$_{\text{w SLAPS}}$ [7] | 60.65±0.65 | 55.47±1.62 |
| GraphRTA | 62.33±1.53 | 56.91±2.50 |

Table 9: MMD comparison before and after graph reprogramming.

| Methods | A→C | A→D | C→A | C→D | D→A | D→C |
|---------|------|------|------|------|------|------|
| Before | 0.0381 | 0.0402 | 0.0368 | 0.0399 | 0.0371 | 0.0378 |
| After | 0.3520 | 0.3932 | 0.5081 | 0.4682 | 0.3608 | 0.4477 |

Table 10: LLM integration for open-set detection.

| Methods | Arxiv-I→Arxiv-II | Arxiv-I→Arxiv-III | Arxiv-II→Arxiv-III |
|---------|------------------|-------------------|--------------------|
| A2GNN | 45.00±0.24 | 43.14±0.18 | 45.18±0.17 |
| A2GNN + LLM-explanations | 46.40±0.45 | 43.34±0.18 | 48.81±0.49 |
| SDA | 42.60±0.15 | 39.44±0.21 | 46.03±0.18 |
| SDA + LLM-explanations | 44.10±1.75 | 40.55±0.10 | 48.50±0.69 |
| GraphRTA | 50.79±2.79 | 46.25±0.40 | 48.42±1.94 |
| GraphRTA + LLM-explanations | 51.56±1.23 | 47.59±0.48 | 50.18±0.79 |

**Comparison with Alternative Graph Reprogramming Strategies.** We also evaluate two alternative graph reprogramming approaches, SUBLIME [31] and SLAPS [7], both of which employ self-supervised learning techniques. The results, summarized in Table 8, reveal that our approach outperforms these strategies. SUBLIME relies on GNN-based node similarity learning followed by KNN-based sparsification to generate a sparse adjacency matrix. This dependence on nearest neighbors limits its flexibility and performance. SLAPS, on the other hand, employs a denoising autoencoder loss for graph reconstruction, which introduces constraints that may not generalize well across different datasets. By avoiding such dependencies, our method achieves superior versatility and effectiveness, as demonstrated in the experiments.

**How Does Graph Reprogramming Affect the Feature Distribution of Unknown Classes?** We measure the distributional divergence between known and unknown class features using the Maximum Mean Discrepancy (MMD) metric. This analysis provides a quantitative assessment of whether our approach enhances the separability of known and unknown samples in the latent space. The results shown in Table 9 indicate that graph reprogramming significantly increases the distributional distance between the two groups after adaptation, suggesting that it effectively promotes clearer feature separation and mitigates overlap between known and unknown domains.

## C  Potential of LLM Integration for Unknown Class Detection

Integrating large language models (LLMs) into open-set graph domain adaptation could offer promising opportunities, particularly for datasets where nodes are associated with rich textual information. To explore this direction, we investigate the impact of LLM-generated semantic explanations on unknown class detection. Specifically, we adopt the prompt strategy from TAPE [15] to generate semantic explanations for each node in the ogbn-arxiv dataset. For each node, we construct textual features by concatenating the title, abstract, and the corresponding LLM-generated explanation, thereby enriching the semantic content available to the model.

We then evaluate the effectiveness of these enhanced node features using three representative open-set graph adaptation models. Each experiment is conducted five times with different random seeds, and we report the average H-score along with the standard deviation to ensure statistical reliability. The results, summarized in Table 10, show that incorporating LLM-generated explanations consistently improves performance across all evaluated models. This improvement highlights that LLMs can provide complementary semantic signals that help distinguish between known and unknown classes, thereby enhancing the robustness of open-set graph domain adaptation. Overall, these findings suggest that leveraging LLM-generated semantic knowledge can substantially benefit open-set scenarios, especially in text-rich graph datasets. We consider this a promising direction for future research.

# D Complexity Analysis

Let $\mathcal{G}_s$ denote the source graph consisting of $n_s$ nodes and $e_s$ edges, $\mathcal{G}_t$ represent the target graph with $n_t$ nodes and $e_t$ edges. Assume that the node representation dimension is $d$ and the graph neural network has $L$ layers. Then, the time complexity associated with encoding the feature representations of both the source and target graphs is given by $\mathcal{O}(Ld^2(n_s + n_t) + Ld(e_s + e_t))$. For the model reprogramming phase, the complexity of sorting the gradient magnitudes is $\mathcal{O}(Ld\log(d))$. In the context of graph reprogramming, the complexity for transforming node features is $\mathcal{O}(n_t d)$, while the refinement of the graph structure takes a complexity of $\mathcal{O}(e_t)$. Furthermore, additional computation arises from the posterior inference process, which is characterized by a complexity of $\mathcal{O}(n_t T)$, where $T$ denotes the number of iterations in the Expectation-Maximization (EM) procedure, assumed to be a constant. Thus, the overall computational complexity of the proposed framework falls within the same order of magnitude as existing methods.

Finally, to enhance clarity about our training procedure, we present a comprehensive step-by-step outline in Algorithm 1.

---

**Algorithm 1** GraphRTA's Training Strategy

---

1: **Input:** Given a labeled source graph $\mathcal{G}_s$ and an unlabeled target graph $\mathcal{G}_t$, graph neural network $\Phi = f_w \circ g_\phi$
2: **Output:** Target graph predictions $\mathbf{Y}_t \in \mathbb{R}^{n_t \times |\mathcal{C}_s| + 1}$
3: Randomly initialize weights of $f_w$ and $g_\phi$
4: **while** not reached the maximum epochs **do**
5:   **for** batch data from source and target graph **do**
6:     Fix the parameters of graph reprogramming
7:     Conduct model reprogramming with Eq.( 2)
8:   **end for**
9:   Update model as reprogrammed GNN
10:   **for** batch data from source and target graph **do**
11:     Fix the parameters of model reprogramming
12:     Conduct graph reprogramming with Eq.( 4)
13:   **end for**
14:   Update target graph as reprogrammed graph
15: **end while**
16: Compute $\mathbf{Y}_t \in \mathbb{R}^{n_t \times |\mathcal{C}_s| + 1}$ with graph neural network

---

