# OpenReview forum: "Towards Unsupervised Open-Set Graph Domain Adaptation via Dual Reprogramming"
_NeurIPS.cc/2025/Conference — NeurIPS 2025 poster_

### Official Review · Reviewer_2LyR · 2025-06-25

**Clarity:** 3
**Significance:** 2
**Originality:** 2
**Rating:** 4
**Confidence:** 3

**Summary:**

This paper addresses the problem of unsupervised open-set graph domain adaptation, where the goal is not only to correctly classify target nodes belonging to known classes but also to detect previously unseen node types as unknown. The authors propose a framework called GraphRTA, which performs reprogramming on both the graph and model levels. Specifically, they modify the target graph structure and node features to improve the separation of known and unknown classes. Domain-specific parameters are pruned to reduce bias toward the source graph while retaining transferable patterns. The classifier is extended with an additional dimension to represent the unknown class.

**Questions:**

Q1: How is the value of the budget B determined?

Q2: How is the learning of \delta A achieved in practice?

Additionally, my current evaluation assumes that the authors will successfully address the concerns and confusions mentioned in the weaknesses section, so I would appreciate it if the authors could also respond to the questions raised there.

**Ethical Concerns:**

["NO or VERY MINOR ethics concerns only"]

**Final Justification:**

The authors’ rebuttal addressed most of my initial concerns, particularly regarding methodological details and hyperparameter configurations, and provided helpful clarifications. These clarifications should be incorporated into the paper to improve clarity for a broader audience. I also note that the Beta mixture model, a key component of the method, closely resembles techniques in prior work. The paper should more explicitly acknowledge related literature and clarify the originality of this component to avoid potential misinterpretation. Some reviewer has raised concerns about the technical novelty, given that the core techniques are adaptations of existing methods from non-graph open-set domains. While I am familiar with non-graph DA techniques, I am less familiar with graph-based DA approaches and will therefore take a conservative stance regarding the novelty of this work. Taking all these factors into account, I will maintain my rating and confidence in my review.

**Limitations:**

Yes.

**Quality:**

3

**Strengths And Weaknesses:**

Strengths:

1. The paper proposes a new solution for open-set domain adaptation (OSDA) in graph neural networks, explicitly distinguishing between known and unknown node classes.

2. The overall writing is generally clear and fluent, and the algorithm is logically presented.

3. Experimental results demonstrate improved performance compared to existing methods.


Weaknesses:

1. Some parts of the paper still remain unclear, which makes it difficult to fully understand the core ideas. For example, in the second paragraph of Section 3.3, the use of M as a mask in the node representation is not well explained. It appears to be a learnable parameter, yet the following text mentions that weights with small gradients with respect to M are masked. This suggests M might instead be derived from another process. The explanation needs clarification.

2. In the same section, the hyperparameter \rho is mentioned, but its definition and role in the model are not clearly described. Additionally, a dynamic adaptive threshold is referenced but not sufficiently explained. Since this is claimed as a key contribution, it should be fully justified and elaborated.

3. On line 200, it is unclear how \delta A is computed. If it is a learnable parameter, how is the binary matrix learned? Also, the choice of the budget B is not discussed. This seems like an important hyperparameter that could significantly impact performance, so its value, how to choose, and default setting should be explained in more detail.

4. The paper employs a Beta mixture model, which appears nearly identical to the one proposed in "Unknown-Aware Domain Adversarial Learning for Open-Set Domain Adaptation". Although the authors cite this work in the introductory description at the beginning of the paragraph, they should explicitly acknowledge the similarity in this section, as the techniques are almost the same.

5. While the proposed method shows strong performance in most scenarios in Tables 2 and 3, it would be helpful to discuss the cases where the method does not achieve the best results, and explore possible reasons why.

---

> ### Author Rebuttal · Authors · 2025-07-31
>
> We sincerely thank the Reviewer 2LyR for the thorough and insightful feedback. Our detailed responses to your concerns are listed as follows:
>
> ***
>
> *Q1. In the second paragraph of Section 3.3, the use of $M$ as a mask in the node representation is not well explained. The explanation needs clarification.*
>
> A1. Thank you for your helpful feedback. In our method, $M$ is a learnable parameter, initialized randomly and updated during training. It is applied as a soft mask on graph neural network weights to enable domain invariant feature learning during model reprogramming. The sentence referencing weights with small gradients with respect to $M$ being masked was intended to illustrate how uninformative components are implicitly suppressed during training via gradient-based optimization, not that $M$ is derived from another process. We will revise this paragraph in the final version to make the role of $M$ as a learnable soft mask more explicit and remove any misleading phrasing that may have caused confusion.
>
> ***
>
> *Q2. In the same section, the hyperparameter $\rho$ is mentioned, but its definition and role in the model are not clearly described. Additionally, a dynamic adaptive threshold is referenced but not sufficiently explained. Since this is claimed as a key contribution, it should be fully justified and elaborated.*
>
> A2. Thank you for pointing this out. The hyperparameter $\rho$ is used to control the sparsity of the learned mask $M$ in the model reprogramming process. Specifically, it determines the proportion of weights to be retained, allowing the model to focus on the most informative components of the representation while suppressing less relevant ones. We set the lowest $\rho$ percent of gradient values in $M$ to zero, leaving the remaining elements at one. These sparse masks are then applied to prune the weight matrix $W$, resulting in a reprogrammed sparse model that emphasizes informative regions of the feature space.
>
> For dynamic threshold, we mean we adopt a logit augmentation strategy, where we append an additional logit dimension to represent the "unknown" class. This augmented logit vector is then passed through a softmax layer, allowing the model to assign a probability distribution over both known and unknown classes. The final prediction is simply the class with the highest posterior probability. The "dynamic threshold" referenced in the text refers not to an explicit numerical threshold, but to the model's ability to adaptively learn the decision boundary between known and unknown classes based on the input node representation $z$ and the augmented classifier outputs. This formulation removes the need for hand-tuned thresholds and makes the open-set detection process fully data-driven and end-to-end trainable. We will revise the wording in the final version to avoid ambiguity and better reflect this mechanism.
>
> ***
>
> *Q3. On line 200, it is unclear how $\Delta{A}$ is computed. If it is a learnable parameter, how is the binary matrix learned? Also, the choice of the budget $B$ is not discussed.*
>
> A3. Thank you for your thoughtful question. To maintain a discrete and sparse adjacency structure, we treat each entry in $\Delta{A}$ as a Bernoulli random variable, where the probability of an edge being present is learned. During training, we sample the learned graph structure from these Bernoulli distributions, allowing the model to adaptively reprogram the input graph while keeping the structure binary. This formulation enables stochastic but learnable structure refinement in a principled manner.
>
> Regarding the budget $B$, it is used to constrain the total number of edges that can be added or modified in $A$, which helps control overfitting and keeps the reprogramming meaningful. In practice, we set $B$ as a fixed percentage (e.g., 20%) of the total possible edges and have provided an **ablation study in Figure 2(b)** to analyze its impact. We will add a more detailed explanation of this process, along with the default setting in the final version of the paper.
>
> ***
>
> *Q4. The paper employs a Beta mixture model, which appears nearly identical to the one proposed.*
>
> A4. Thank you for your feedback. We agree that the Beta mixture model we adopt is closely aligned with the previous work. We would like to clarify that our primary contribution is not in designing a new loss function, but in introducing a novel learning paradigm, i.e., dual reprogramming for open-set graph domain adaptation, which integrates model and graph reprogramming to address challenges specific to graph-structured data. The Beta mixture model is used as a standard component within our framework to support open-set recognition, consistent with prior work.
>
> ***
>
> *Q5. While the proposed method shows strong performance in most scenarios in Tables 2 and 3, it would be helpful to discuss the cases where the method does not achieve the best results and explore possible reasons why.*
>
> A5. Thank you for your insightful comment. We appreciate your suggestion to analyze the cases where our method does not achieve the best performance. Upon reviewing the results in Tables 2 and 3, we note that in a few scenarios, our method performs slightly below certain closed-set baselines. These cases often occur in settings where the known class distribution is very similar with unknown class distributions across the two domains. We compute the Maximum Mean Discrepancy (MMD) between known and unknown classes within each domain and then calculate their MMD gap between the source and target domains. We find that DBLPv7->ACMv9 (0.003) and Texas->Cornell (0.024) have the smallest values among all the scenarios. In this case, closed-set models leverage the full target space for alignment without needing to separate unknown classes, resulting in a performance improvement. We will include this analysis and a brief discussion of these observations in the final version to help readers better understand our approach.
>
> ***
>
> *Q6. How is the value of the budget $B$ determined?*
>
> A6. Thanks for pointing this out. The budget $B$ is treated as a hyperparameter that controls the sparsity of the reprogrammed graph structure. In our experiments, we set $B$ to a fixed proportion (e.g., 20%) of the total possible edges. We further analyze the sensitivity of the model to different values of $B$ through an ablation study, as shown in Figure 2(b).
>
> ***
>
> *Q7. How is the learning of $\Delta{A}$ achieved in practice?*
>
> A7. Thanks for pointing this out. To learn $\Delta A$ in practice, we treat each entry in $\Delta A$ as a Bernoulli random variable, where the probability of an edge being present is learned. During training, we sample the learned graph structure from these Bernoulli distributions, allowing the model to adaptively reprogram the input graph while keeping the structure binary. The learning process is further constrained by a budget $B$ via projected gradient descent, which limits the number of added or modified edges. This formulation enables stochastic but learnable structure refinement in a principled manner. We will add a more detailed description in the final version.

---

> > ### Comment · Reviewer_2LyR · 2025-08-05
> >
> > I would like to thank the authors for the effort taken to address the concerns and questions raised in my initial review. Most of my comments were related to the detailed steps of the proposed method, as well as the configuration and tuning of hyperparameters. The rebuttal has helped clarify some of these points. Being that said, I would strongly recommend that the authors revise the paper to incorporate these clarifications. It is possible that I may have overlooked some of the details during my initial reading, but the absence of such information can make the paper difficult to follow for a broader audience.
> >
> > I also have a follow-up point regarding the use of the Beta mixture model. This component appears to play an important role in the method and closely resembles techniques used in prior work. While the authors mention that this model is adopted as a standard component, I believe the paper should have a more thorough discussion of relevant prior literature and a clearer attribution of credit. As currently written, the phrasing may give the impression that this component is a novel contribution, which may be misleading to readers. It would be highly encouraged to appropriately acknowledge related work and clarify the originality of their contribution.

---

> > > ### Author Response · Authors · 2025-08-06
> > >
> > > We sincerely thank the reviewer for the feedback. We agree that the clarity of methodological details and hyperparameter configurations is essential for broader accessibility. In the final version, we will revise the manuscript to explicitly incorporate the explanations and clarifications provided in our rebuttal, including additional details on model steps, hyperparameter settings to improve readability and reproducibility.
> > >
> > > Regarding the loss function of Beta mixture model, we agree that clearer attribution and discussion of related literature are necessary. In the revised manuscript, we will update Section 3.5 to explicitly acknowledge this prior work, clarify the role of the Beta mixture model within our framework, and distinguish our contributions from existing methods more transparently.
> > >
> > > Thank you again for the valuable feedback, which helps improve the clarity and completeness of our work.

---

> ### Comment · Area_Chair_Hbyw · 2025-08-05
> **Author-Reviewer Discussion Reminder**
>
> Dear Reviewer 2LyR,
>
> As the deadline for author-reviewer discussion is approaching, could you please check the authors' rebuttal and post your response?
>
> Thank you!
>
> Best,
>
> AC

---

### Official Review · Reviewer_ny1J · 2025-06-29

**Clarity:** 3
**Significance:** 4
**Originality:** 3
**Rating:** 5
**Confidence:** 5

**Summary:**

This paper studies unsupervised open-set graph domain adaptation. The authors claim that the target domain might include classes not in the source domain in the real-world scenarios. To address this issue, the authors propose to not only classify target nodes into the known classes but also classify unseen node types into the unknown class. Specifically, a novel framework is proposed to conduct reprogramming on both the graph and model sides. For model reprogramming, the authors prune domain specific parameters to reduce bias. For graph reprogramming, the authors reprogram the graphs to facilitate better separation of known and unknown classes. Results on several public datasets show that the proposed model can outperform recent baselines with different gains.

**Questions:**

1.	The authors should provide a more detailed introduction to the datasets used in the experiments. For example, it would be helpful to clarify how open-set classes are selected within each dataset.
2.	One of the recent baselines appears to be missing from the discussion and experimental comparison. Including this baseline would provide a more complete and fair evaluation of the proposed approach. Please refer to reference [1].
3.	Additional ablation studies would strengthen the paper by providing deeper insights into the effectiveness of the proposed method. For example, it would be helpful to explore alternative strategies for open-set detection—such as using an entropy-based threshold—and compare their performance with the proposed approach.
4.	Some of the ablation studies currently placed in the appendix are crucial for understanding the core contributions of the paper and should be moved to the main content.
References:
[1] Shen X, Chen Z, Pan S, et al. Open-Set Cross-Network Node Classification via Unknown-Excluded Adversarial Graph Domain Alignment[C]//Proceedings of the AAAI Conference on Artificial Intelligence. 2025, 39(19): 20398-20408.

**Ethical Concerns:**

["NO or VERY MINOR ethics concerns only"]

**Final Justification:**

I appreciate the rebuttal, and it addresses my concerns regarding the usefulness of the experiment results. Given these considerations, I will raise 4 points to 5. After seeing the discussion in the rebuttal, I strongly support accepting this paper.

**Limitations:**

Yes

**Quality:**

3

**Strengths And Weaknesses:**

Pros:
1.	This paper investigates open-set graph domain adaptation, which is interesting and less explored in the community.
2.	Different types of datasets (citation, ogbn-arxiv and WebKB) are utilized for evaluation.
3.	Comprehensive results demonstrate that its performance is promising.
Cons:
1.	The authors should provide a more detailed introduction to the datasets.
2.	One of the recent baselines is not discussed and compared.
3.	More ablation studies should be given. What if we use entropy threshold for open-set detection?

---

> ### Author Rebuttal · Authors · 2025-07-31
>
> We greatly appreciate Reviewer ny1J for the insightful review to help us improve the paper. We address the reviewer’s concerns as follows:
>
> ***
>
> *Q1. The authors should provide a more detailed introduction to the datasets used in the experiments. For example, it would be helpful to clarify how open-set classes are selected within each dataset.*
>
> A1. Thank you for your suggestion. In our experiments, we follow the common practice used in prior works such as SDA and G2Pxy by selecting the first several classes (based on label indices) as known classes, and the remaining classes as open-set (unknown) classes. We will provide a more detailed description of this setup in the final version of the paper to improve clarity.
>
> ***
>
> *Q2. One of the recent baselines appears to be missing from the discussion and experimental comparison.*
>
> A2. Thank you for bringing this work to our attention. The UAGA method you mentioned is a two-stage framework. It first separates known and unknown classes by training a graph neural network, and then it applies unknown-excluded adversarial domain alignment across different classes. Notably, these operations are performed entirely in the representation space. In contrast, our proposed GraphRTA introduces model reprogramming and graph reprogramming from dual perspectives, which enhances generalization capabilities by explicitly modeling the model space and the graph space. Following your suggestion, we have compared UAGA with GraphRTA on citation datasets. The results of H-score are as follows:
>
> | Methods | A->C | A->D | C->A | C->D | D->A | D->C |
> |:--:|:--:|:--:|:--:|:--:|:--:|:--:|
> | UAGA | 61.34$\pm$1.16 | **67.50$\pm$2.95** | 60.59$\pm$5.95 |64.81$\pm$4.50 | 55.73$\pm$3.98 | 52.16$\pm$2.18 |
> | GraphRTA | **66.33$\pm$1.69** | 64.42$\pm$1.10 | **62.89$\pm$2.46** | **65.99$\pm$1.87** | **59.41$\pm$2.22** | **62.33$\pm$1.53** |
>
> As can be seen, our method demonstrates superior performance over the baseline in 5 out of 6 evaluated scenarios, indicating strong generalization and robustness.
>
> ***
>
> *Q3. It would be helpful to explore alternative strategies for open-set detection—such as using an entropy-based threshold—and compare their performance with the proposed approach.*
>
> A3. Thank you for the insightful suggestion. We agree that exploring alternative open-set detection strategies provides valuable perspective. Based on your suggestion, we implemented an entropy-based thresholding variant and compared its performance with our proposed approach. The results of H-score show that our approach consistently outperforms it across all settings.
>
> | Methods | A->C | A->D | C->A | C->D | D->A | D->C |
> |:--:|:--:|:--:|:--:|:--:|:--:|:--:|
> | GraphRTA-threshold | 63.07$\pm$0.13 | 61.21$\pm$1.07 | 61.00$\pm$1.52 |63.89$\pm$1.35 | 58.88$\pm$0.72 | 59.73$\pm$1.10 |
> | GraphRTA | **66.33$\pm$1.69** | **64.42$\pm$1.10** | **62.89$\pm$2.46** | **65.99$\pm$1.87** | **59.41$\pm$2.22** | **62.33$\pm$1.53** |
>
> We will include these results and discussion in the final version of the paper to provide a more comprehensive evaluation.
>
> ***
>
> *Q4. Some of the ablation studies currently placed in the appendix are crucial for understanding the core contributions of the paper and should be moved to the main content.*
>
> A4. Thank you for your helpful feedback. We agree that some of the ablation studies (e.g., Table 5 and Table 6) are important for understanding the core contributions of our work. In the final version of the paper, we will move these ablation results from the appendix to the main text to improve the clarity of our contributions.

---

> > ### Comment · Reviewer_ny1J · 2025-08-06
> >
> > I appreciate the rebuttal, and it addresses my concerns regarding the usefulness of the experiment results. Given these considerations, I will raise 4 points to 5. After seeing the discussion in the rebuttal, I strongly support accepting this paper.

---

> > > ### Author Response · Authors · 2025-08-06
> > >
> > > Thank you for your support. We will incorporate these experiments into the final version.

---

> ### Comment · Area_Chair_Hbyw · 2025-08-05
> **Author-Reviewer Discussion Reminder**
>
> Dear Reviewer ny1J,
>
> As the deadline for author-reviewer discussion is approaching, could you please check the authors' rebuttal and post your response?
>
> Thank you!
>
> Best,
>
> AC

---

### Official Review · Reviewer_bwmJ · 2025-07-02

**Clarity:** 3
**Significance:** 2
**Originality:** 2
**Rating:** 3
**Confidence:** 4

**Summary:**

This paper addresses the problem of unsupervised open-set graph domain adaptation (UOSGDA), where the target graph contains classes unseen in the source domain. The authors propose GraphRTA, a dual-reprogramming framework that modifies both graph structure/node features ("graph reprogramming") and prunes domain-specific model parameters ("model reprogramming"). Key innovations include: (1) Joint graph and model adaptation to separate known/unknown classes; (2) An extended classifier with an additional "unknown" dimension to avoid manual thresholding. Experiments on public datasets demonstrate superiority over GNNs, closed-set GDA, and open-set graph learning baselines.

**Questions:**

1.Potential of LLM Integration: Given datasets with text-describable nodes (e.g., Citation networks), have you considered leveraging LLMs for unknown class detection? LLMs could generate semantic explanations for classification decisions and handle the open-set classification.

2.There is an inconsistency in reported baselines: Table 4 cites GCN HS metric as 56.91%, while Table 2 lists as 59.41%. Please verify and correct all numerical results.

**Ethical Concerns:**

["NO or VERY MINOR ethics concerns only"]

**Limitations:**

Yes

**Quality:**

2

**Strengths And Weaknesses:**

Quality:
Strength: Experimental design is rigorous, covering diverse baseline types (GNNs, closed-set GDA, open-set methods) across multiple datasets.
Weakness: Model complexity raises concerns. The combination of adversarial training, multiple loss functions, and dual reprogramming may lead to training instability and convergence difficulties.

Clarity:
Strength: The manuscript is well-structured and clearly written, with logically motivated components.

Significance and Originality:
Weakness: While UOSGDA is a relevant problem, the technical novelty is limited. Core techniques (entropy-based known/unknown separation, adversarial domain alignment) are adaptations of existing methods from non-graph open-set domains. The incremental contribution does not sufficiently advance the state-of-the-art.

---

> ### Author Rebuttal · Authors · 2025-07-31
>
> We sincerely thank Reviewer bwmJ for the constructive feedback. We address the reviewer’s concerns as follows.
>
> ***
>
> *Q1. Model complexity raises concerns. The combination of adversarial training, multiple loss functions, and dual reprogramming may lead to training instability and convergence difficulties.*
>
> A1. Thank you for raising this point. For model complexity, we have provided a detailed model complexity analysis in the **Appendix D**, which shows that the overall computational overhead remains manageable and comparable to existing domain adaptation methods. For training instability, we mitigate potential instability by using the gradient reversal layer (GRL), a well-established technique in adversarial training that enables stable joint optimization without requiring alternating updates. In our experiments, we did not observe instability or convergence issues during training across all datasets. We will include additional training details and convergence plots in the final version to demonstrate the stability of our method.
>
> ***
>
> *Q2. Core techniques (entropy-based known/unknown separation, adversarial domain alignment) are adaptations of existing methods from non-graph open-set domains. The incremental contribution does not sufficiently advance the state-of-the-art.*
>
> A2. Thank you for your feedback. We believe our proposed dual reprogramming provides a new and meaningful direction for open-set graph domain adaptation that goes beyond adapting existing ideas. By reprogramming the model and the graph from dual perspectives, the method effectively reshapes the learning objective to focus on domain-invariant and known-class-relevant information. This is a novel strategy that contributes to the method’s strong performance under open-set conditions. To summarize, we would like to emphasize our contributions as follows:
>
> (1) For **problem**, we explore the challenging yet practical problem of unsupervised open-set graph domain adaptation, which remains relatively uninvestigated within the graph community.
> (2) For **model**, we propose a dual reprogramming framework named GraphRTA that jointly reprograms the model and the graph structure to better handle the open-set setting in graphs. This is fundamentally different from conventional methods that operate solely in the representation space.
> (3) For **evaluation**, we conduct extensive comparisons with 14 baselines to validate the effectiveness of our proposed GraphRTA. The baselines encompass various categories, such as no-adaptation methods, closed-set graph domain adaptation, and open-set graph domain adaptation methods.
>
> We believe these contributions offer a substantive advancement in the field of open-set graph domain adaptation.
>
> ***
>
> *Q3. Potential of LLM Integration: Given datasets with text-describable nodes (e.g., Citation networks), have you considered leveraging LLMs for unknown class detection? LLMs could generate semantic explanations for classification decisions and handle the open-set classification.*
>
> A3. Thank you for the insightful suggestion. We agree that integrating large language models (LLMs) into open-set graph domain adaptation holds great potential, especially for datasets with text-describable nodes. To explore the impact of LLM generated semantic explanations on unknown class detection, we follow the prompt strategy from TAPE [1] to generate semantic explanations for each node in the **ogbn-arxiv dataset**. The node features are then constructed by combining the title, abstract, and the LLM-generated explanation. We evaluate the effectiveness of these enhanced features using three models. Each experiment is run 5 times, and we report the average H-score along with the standard deviation as follows:
>
> | Methods | Arxiv I->Arxiv II | Arxiv I->Arxiv III | Arxiv II->Arxiv III |
> |:--:|:--:|:--:|:--:|
> | A2GNN | 45.00$\pm$0.24 | 43.14$\pm$0.18 | 45.18$\pm$0.17 |
> | A2GNN + LLM-explanations | **46.40$\pm$0.45** | **43.34$\pm$0.18** | **48.81$\pm$0.49** |
> | SDA | 42.60$\pm$0.15 | 39.44$\pm$0.21 | 46.03$\pm$0.18 |
> | SDA + LLM-explanations | **44.10$\pm$1.75** | **40.55$\pm$0.10** | **48.50$\pm$0.69** |
> | GraphRTA | 50.79$\pm$2.79 | 46.25$\pm$0.40 | 48.42$\pm$1.94 |
> | GraphRTA + LLM-explanations | **51.56$\pm$1.23** | **47.59$\pm$0.48** | **50.18$\pm$0.79** |
>
> As shown in the table, incorporating LLM-generated explanations leads to performance improvements across the evaluated models. This suggests that LLMs can enhance the models and provide additional semantic signals beneficial for open-set classification. We consider this a promising direction for future research and will include a brief discussion of this potential extension in the final version of the paper.
>
> Reference:
>
> [1] He X, Bresson X, Laurent T, et al. Harnessing Explanations: LLM-to-LM Interpreter for Enhanced Text-Attributed Graph Representation Learning[C]// ICLR 2024.
>
> ***
>
> *Q4. There is an inconsistency in reported baselines: Table 4 cites GCN HS metric as 56.91%, while Table 2 lists as 59.41%. Please verify and correct all numerical results.*
>
> A4. Thank you for pointing this out. We have carefully reviewed all numerical results and confirm that the discrepancy is due to a typo in Table 4, where the accuracy value (56.91%) was mistakenly placed in the H-score for GCN backbone. The correct H-score is 59.41%, as reported in Table 2. We have verified all other results, and this is the only such error. The correction will be made in the final version.

---

> ### Comment · Area_Chair_Hbyw · 2025-08-05
>
> Dear Reviewer bwmJ,
>
> As the deadline for author-reviewer discussion is approaching, could you please check the authors' rebuttal and post your response?
>
> Thank you!
>
> Best,
>
> AC

---

> > ### Comment · Area_Chair_Hbyw · 2025-08-07
> >
> > Dear Reviewer,
> >
> > Could you please check the authors' rebuttal and post your response?
> >
> > The new policy requires AC to flag insufficient review this year, including the non-participation in author-reviewer discussions.
> >
> > Thanks,
> >
> > AC

---

### Official Review · Reviewer_s6Bn · 2025-07-02

**Clarity:** 3
**Significance:** 4
**Originality:** 3
**Rating:** 5
**Confidence:** 4

**Summary:**

This paper addresses the task of unsupervised open-set graph domain adaptation, where the target domain may contain classes do not present in the source domain, an important consideration in real-world applications. To tackle this challenge, the authors propose a novel framework that simultaneously classifies target nodes into known categories while identifying unseen node types as belonging to an unknown class. The approach involves reprogramming at both the model and graph levels. Specifically, model reprogramming is achieved by pruning domain-specific parameters to mitigate bias, while graph reprogramming adjusts graph structures to enhance the separation between known and unknown classes. Experimental results on multiple public datasets demonstrate that the proposed method outperforms recent baselines by varying margins.

**Questions:**

1. The authors should discuss and compare with the following AAAI 2025 paper [1].
[1] Shen X, Chen Z, Pan S, et al. Open-Set Cross-Network Node Classification via Unknown-Excluded Adversarial Graph Domain Alignment, Proceedings of the AAAI Conference on Artificial Intelligence. 2025, 39(19): 20398-20408.
2. Incorporating further ablation studies could enhance the paper. Specifically, the effect of varying the number of known classes—from 2 up to 4—on the model’s performance remains unexplored. Investigating this aspect would provide valuable insight into the model’s robustness under different complexities.
3. The organization of the paper could be further improved by simplifying Section 3.5. Streamlining this section would enhance clarity and help readers follow the key concepts more easily, making the overall presentation more coherent and accessible.
4. How does graph reprogramming affect the feature distribution of unknown classes? A detailed analysis of this effect would help clarify whether the approach effectively enhances the distinction between known and unknown classes.

**Ethical Concerns:**

["NO or VERY MINOR ethics concerns only"]

**Final Justification:**

The score I have provided accurately reflects my current positive evaluation of the paper, and I will therefore maintain my current score.

**Limitations:**

Yes

**Quality:**

3

**Strengths And Weaknesses:**

# Strengths:
1.	This paper is well written and easy to understand.
2.	This paper investigates an important and interesting problem, i.e., open-set graph domain adaptation.
3.	The model’s performance is pretty good compared with existing methods.
# Weaknesses:
1.	The authors fail to discuss and compare with a recent relevant baseline.
2.	Additional ablation studies would strengthen the paper. For instance, it remains unclear how varying the number of known classes impacts the model’s performance.
3.	The description of the training procedure is too long, which could be simplified.

---

> ### Author Rebuttal · Authors · 2025-07-31
>
> We greatly appreciate Reviewer s6Bn for the insightful review to help us improve the paper. We address the reviewer’s concerns as follows:
>
> ***
>
> *Q1. The authors should discuss and compare with the following AAAI 2025 paper.*
>
> A1. Thank you for bringing this work to our attention. The UAGA method you mentioned is a two-stage framework. It first separates known and unknown classes by training a graph neural network, and then it applies unknown-excluded adversarial domain alignment across different classes. Notably, these operations are performed entirely in the representation space. In contrast, our proposed GraphRTA introduces model reprogramming and graph reprogramming from dual perspectives, which enhances generalization capabilities by explicitly modeling the model space and the graph space. Following your suggestion, we have compared UAGA with GraphRTA on citation datasets. The results of H-score are as follows:
>
> | Methods | A->C | A->D | C->A | C->D | D->A | D->C |
> |:--:|:--:|:--:|:--:|:--:|:--:|:--:|
> | UAGA | 61.34$\pm$1.16 | **67.50$\pm$2.95** | 60.59$\pm$5.95 |64.81$\pm$4.50 | 55.73$\pm$3.98 | 52.16$\pm$2.18 |
> | GraphRTA | **66.33$\pm$1.69** | 64.42$\pm$1.10 | **62.89$\pm$2.46** | **65.99$\pm$1.87** | **59.41$\pm$2.22** | **62.33$\pm$1.53** |
>
> As can be seen, our method demonstrates superior performance over the baseline in 5 out of 6 evaluated scenarios, indicating strong generalization and robustness.
>
> ***
>
> *Q2. The effect of varying the number of known classes—from 2 up to 4—on the model’s performance remains unexplored.*
>
> A2. Thanks for this suggestion. Following your suggestion, we have conducted additional ablation studies to evaluate the impact of varying the number of known classes (from 2 to 4) on model performance. The results of H-score on citation dataset are presented below:
>
> | Methods | A->C | A->D | C->A | C->D | D->A | D->C |
> |:--:|:--:|:--:|:--:|:--:|:--:|:--:|
> | $C_{kwn}$=2 | 53.39$\pm$1.87 | 45.29$\pm$3.46 | 41.10$\pm$2.63 | 41.54$\pm$2.92 | 44.48$\pm$2.86 | 54.18$\pm$4.41 |
> | $C_{kwn}$=3 | 66.33$\pm$1.69 | 64.42$\pm$1.10 | 62.89$\pm$2.46 | 65.99$\pm$1.87 | 59.41$\pm$2.22 | 62.33$\pm$1.53 |
> | $C_{kwn}$=4 | 65.74$\pm$0.54 | 63.28$\pm$0.69 | 61.63$\pm$0.64 | 67.20$\pm$0.63 | 57.12$\pm$0.35 | 59.82$\pm$0.69 |
>
> When the number of known classes is very small (e.g., only 2 known classes), open-set models have less supervision to guide feature alignment and separation. In contrast, when enough known classes is available, our model becomes more robust and less sensitive is to the number of unknown classes.
>
> ***
>
> *Q3. The organization of the paper could be further improved by simplifying Section 3.5.*
>
> A3. Thank you for the constructive feedback regarding the organization of the paper. We agree that simplifying Section 3.5 would improve clarity and overall coherence. We will revise and streamline this section in the final version to enhance readability and make the key concepts more accessible.
>
> ***
>
> *Q4. How does graph reprogramming affect the feature distribution of unknown classes?*
>
> A4. Thank you for your insightful comment. To analyze how graph reprogramming affects the feature distribution of unknown classes, we measure the distributional divergence between known and unknown classes using Maximum Mean Discrepancy (MMD). This allows us to quantitatively assess whether our approach enhances the separation between these two groups. The results are as follows:
>
> | Methods | A->C | A->D | C->A | C->D | D->A | D->C |
> |:--:|:--:|:--:|:--:|:--:|:--:|:--:|
> | Before | 0.0381 | 0.0402 | 0.0368 | 0.0399 | 0.0371 | 0.0378 |
> | After | 0.3520 | 0.3932 | 0.5081 | 0.4682 | 0.3608 | 0.4477 |
>
> The results indicate that graph reprogramming increases the distributional distance between known and unknown classes after graph reprogramming, suggesting improved feature separation. We will include this analysis in the final version of the paper.

---

### Note · Authors · 2025-08-13

We sincerely thank all reviewers for taking the time to review our paper and for providing detailed, constructive feedback. We are encouraged that our work has been recognized as addressing an important and interesting problem (Reviewers s6Bn, ny1J, 2LyR), demonstrating superior experimental performance (Reviewers s6Bn, bwmJ, ny1J, 2LyR), and being well-organized and clearly written (Reviewers s6Bn, bwmJ, ny1J, 2LyR).

During the rebuttal stage, we carefully addressed each concern with detailed, point-by-point responses. After this phase, Reviewers s6Bn and ny1J indicated that their concerns had been fully resolved. Reviewer s6Bn maintained a positive rating, while Reviewer ny1J increased the score by one point.

Reviewer bwmJ raised concerns regarding the evaluation of integrating large language models (LLMs) into open-set graph domain adaptation. In response, we conducted additional experiments on the ogbn-arxiv dataset. The results show that incorporating LLM-generated explanations improves performance across the evaluated models, confirming the potential of this direction. Reviewer 2LyR acknowledged that the rebuttal clarified several points and suggested revising the paper to incorporate these clarifications. We will make these revisions in the final version to further enhance clarity and completeness.

We appreciate the constructive comments from all reviewers and hope that our detailed responses and additional experiments have addressed the concerns raised.

---

### Decision · Program_Chairs · 2025-09-17

**Decision:**

Accept (poster)

**Comment:**

This paper aims to address the unsupervised open-set graph domain adaptation problem, where the goal is not only to correctly classify target nodes belonging to known classes but also to detect previously unseen node types as unknown. To this end, the authors  proposed a dual-reprogramming framework named GraphRTA, which modifies both graph structure/node features ("graph reprogramming") and prunes domain-specific model parameters ("model reprogramming"). Experiments on public datasets demonstrate superiority over GNNs, closed-set GDA, and open-set graph learning baselines.

Reviewers recognized the novelty and technical contributions of this work. This paper studies an important and challenging problem from a unique perspective. The idea of GraphRTA is well motivated and clearly explained. The extensive results also validated the effectiveness of the proposed approach.

Meanwhile, reviewers raised some concerns about missing baselines, ablation studies, potential LLM integration, model complexity, etc. The authors have provided detailed responses with additional results, which have addressed most of these concerns from reviewers. The authors are strongly encouraged to incorporate the new results and discussions into the final version of the paper.